# The Role of Food Security in Mediterranean Diet Adherence Among Adolescents: Findings from the EHDLA Study

**DOI:** 10.3390/foods14030414

**Published:** 2025-01-27

**Authors:** Andrea Aquino-Blanco, Estela Jiménez-López, Desirée Victoria-Montesinos, Héctor Gutiérrez-Espinoza, Jorge Olivares-Arancibia, Rodrigo Yañéz-Sepúlveda, Nerea Martín-Calvo, José Francisco López-Gil

**Affiliations:** 1Health and Social Research Center, Universidad de Castilla-La Mancha, 16071 Cuenca, Spain; aaquinoblanco@gmail.com (A.A.-B.); estela.jimenezlopez@uclm.es (E.J.-L.); 2Centro de Investigación Biomédica en Red de salud Mental, Instituto de Salud Carlos III, 28029 Madrid, Spain; 3Faculty of Pharmacy and Nutrition, UCAM Universidad Católica San Antonio de Murcia, 30107 Murcia, Spain; dvictoria@ucam.edu; 4Faculty of Education, Universidad Autónoma de Chile, Santiago 7500912, Chile; kinehector@gmail.com; 5AFySE Group, Research in Physical Activity and School Health, School of Physical Education, Faculty of Education, Universidad de Las Américas, Santiago 7500975, Chile; jorge.olivares.ar@gmail.com; 6Faculty Education and Social Sciences, Universidad Andres Bello, Viña del Mar 2520000, Chile; rodrigo.yanez.s@unab.cl; 7Department of Preventive Medicine and Public Health, Facultad de Medicina, Universidad de Navarra, 31008 Pamplona, Spain; nmartincalvo@unav.es; 8Instituto de Investigación Sanitaria de Navarra (IdiSNA), 31008 Pamplona, Spain; 9Pathophysiology of Obesity and Nutrition, Centro de Investigación Biomédica en Red, Instituto de Salud Carlos III, 28029 Madrid, Spain; 10One Health Research Group, Universidad de Las Américas, Quito 170124, Ecuador

**Keywords:** socioeconomic status, food insecurity, diet quality, eating healthy, lifestyle, teenagers

## Abstract

Food insecurity is a growing public health concern, particularly among vulnerable groups such as adolescents, and it has been linked to poor nutritional outcomes and increased risk of chronic diseases. The Mediterranean diet (MedDiet) is recognized for its numerous health benefits, yet few studies have explored the association between food insecurity and adherence to the MedDiet among adolescents, especially in Mediterranean regions. Objective: This study aimed to investigate the association between food insecurity and adherence to the MedDiet in a sample of adolescents from Spain. Methods: Data were acquired from 2021–2022 school years from adolescents aged 12–17 years enrolled in different secondary schools in *Valle de Ricote*, Region of Murcia, Spain. Food security was assessed via the Spanish Child Food Security Survey Module (CFSSM-S), and adherence to the MedDiet was assessed via the Mediterranean Diet Quality Index for Children and Adolescents (KIDMED). A total of 836 participants were involved, and statistical analyses were conducted via linear regression models adjusted for lifestyle, anthropometric, and sociodemographic covariates. Results: After adjusting for potential confounders, the mean KIDMED score was 7.0 (95% confidence interval [CI] 6.7 to 7.2) in the food-security group, 6.8 (95% CI 6.3 to 7.3) in the low-security group, and 5.9 (95% CI 4.9 to 6.9) in the very-low-security group. Notable differences were observed between participants with food security and their counterparts with very low food security (*p* = 0.040). Compared with those with food security (57.2%, 95% CI 51.5% to 62.8%), a significantly greater likelihood of having nonoptimal adherence to the MedDiet was identified in adolescents with low food security (61.6%, 95% CI 50.4% to 71.8%) and in those with very low food security (77.4%, 95% CI 54.5% to 90.7%). We identified a significant difference only between adolescents with food security and their peers with very low food security (*p* = 0.036). Conclusions: The findings suggest that very low food security negatively impacts MedDiet adherence in adolescents. Compared with their food-secure peers, adolescents with very low food security showed significantly poorer diet quality. These results highlight the importance of addressing food insecurity through low-cost, high-nutrition programs aimed at improving healthy eating habits, particularly for children and adolescents living in low-food-security households.

## 1. Introduction

The Mediterranean diet (MedDiet) is an eating pattern characterized by high consumption of vegetables, nuts, fruits, complex carbohydrates, legumes, and herbs, and low consumption of red and processed meats. It involves a moderate intake of seafood, fish, white meat, eggs, and dairy products, with olive oil being the main source of added fat [1,2]. Nutritionally, the MedDiet is distinguished by its focus on low-glycemic index carbohydrates, monounsaturated fats, dietary fiber, antioxidants, vegetable proteins, low intake of ultraprocessed foods, and a balanced omega-6/omega-3 polyunsaturated fatty acid ratio [3]. The link between the MedDiet and increased longevity has gained significant attention over the past twenty years because of its various health benefits [4]. Several studies suggest that the MedDiet helps protect against numerous health conditions, including coronary heart disease, diabetes, atherosclerosis, and metabolic syndrome, among others [2,5]. In 2020, the MedDiet was updated to include an additional dimension in the pyramid, with environmental sustainability becoming a central component. This change highlights the importance of reducing red meat consumption while increasing the intake of locally grown, eco-friendly plant-based foods and legumes [2]. The new Pyramid for a sustainable MedDiet by Serra-Majem [2], illustrating these updates, can be found at this link (https://www.mdpi.com/1660-4601/17/23/8758) (accessed on 10 January 2025). This resource promotes the adoption of sustainable food production and consumption practices aimed not only at improving health outcomes but also at mitigating environmental degradation.

Since the 2000s, the economic conditions in numerous European nations have deteriorated, resulting in financial insecurity, social uncertainty, and a notable rise in mortality rates linked to chronic illnesses [6]. In parallel, this economic downturn has intensified a broader nutritional transition marked by declining adherence to the MedDiet. Reduced purchasing power has made nutrient-rich foods such as fresh produce, fish, and olive oil increasingly unaffordable for many households. Consequently, consumers have shifted toward cheaper, calorie-dense, and highly processed foods, resulting in poorer overall dietary quality and heightened health risks [7]. This recent economic crisis and the social disparities associated with childhood obesity have made the price of food a significant barrier to maintaining an adequate MedDiet pattern [8].

The Food and Agriculture Organization (FAO) of the United Nations defines food insecurity as a status where individuals do not have consistent access to adequate, nutritious food essential for healthy development and growth and maintaining an active lifestyle [9]. This lack of access can result from various factors, including economic hardship, supply chain disruptions, and social inequalities [9]. According to a report from the FAO, in 2021, approximately 1.5% of Spanish households suffered from severe food insecurity [10]. As food security status is a health determinant associated with economic, geographic, and social components, its effects on health are multiple, both direct and indirect [11]. Thus, evidence indicates that households experiencing very low food security tend to have poorer overall health, receive fewer preventive medical and dental care services, and face higher rates of conditions such as anemia and mental health disorders [12]. Similarly, a systematic review revealed that very low food security was adversely related to diet quality in the United States and was associated with a low intake of foods rich in nutrients such as vegetables, fruits, and dairy products, which are known to promote a balanced diet and health [13].

Adolescence is a pivotal phase in human development marked by complex and accelerated neurodevelopmental, physical, social, and psychological modifications [14]; therefore, nutritional deficiencies put adolescents at risk of stunted growth and both present and future health issues [15]. During this stage of life, adolescents living in a low-food-security context are likely to experience poor health along with challenges in development, behavior, educational achievement, bullying, and school attendance [16]. Exposure to household food insecurity during this period of life could adversely affect adherence to a healthy dietary style, such as the MedDiet [17].

Before the coronavirus disease 2019 (COVID-19) pandemic, global food insecurity remained a persistent challenge. In 2019, 26.4% of the global population experienced moderate or severe levels of food insecurity [18]. In Europe, food insecurity rates are relatively lower than the global average, but vulnerable groups, including low-income households and marginalized communities, are still significantly affected [19]. In Spain, despite having lower food insecurity rates than developing countries, the situation worsened by 2019, driven by ongoing economic challenges and high unemployment [18]. Since the onset of the pandemic, the prevalence of low or very low food security over the past four years has exceeded pre-COVID-19 levels, resulting in minimal improvement [9]. Therefore, several studies have assessed the associations between low and very low food security and diet quality [20]. For example, studies conducted in Greece [21], Portugal [22], and Lebanon [23] have indicated an inverse association between low and very low food security and adherence to the MedDiet in young adults and adolescents. However, a significant research gap exists concerning Spanish adolescents, despite Spain being an important Mediterranean country. Assessing this population is essential owing to rising socioeconomic inequalities that may affect access to healthy foods such as olive oil, fruits, vegetables, and fish, which are core components of the MedDiet.

Hence, this study aims to investigate the association between food insecurity and adherence to the MedDiet in a sample of adolescents from Spain. Assessing the relationship of food security status in this population could provide valuable insights into dietary shifts, guide policy development, and promote equitable food access. By understanding how food insecurity influences dietary habits, we can identify specific barriers that prevent adolescents from maintaining a healthy diet. This knowledge is crucial for developing targeted interventions that not only address immediate nutritional needs but also foster long-term healthy eating behaviors. Furthermore, such research can highlight the importance of supporting vulnerable populations through comprehensive food policies and community programs, ultimately contributing to the overall well-being and health of adolescents in Spain.

## 2. Materials and Methods

### 2.1. Study Design and Population

This secondary cross-sectional study analyzes information from the Eating Healthy and Daily Life Activities (EHDLA) study, which encompassed a diverse population of teens between 12 and 17 years old from the *Valle de Ricote* in the Region of Murcia (Spain). The data collection took place during the 2021–2022 school year at three distinct secondary educational institutions in the area. Ethical approval for the EDHLA study was granted by two committees: the Ethics Committee of the Albacete University Hospital Complex and Albacete Integrated Care Management (ID 2021-85, approved on 23 November 2021) and the Bioethics Committee of the University of Murcia (ID 2218/2018, approved on 18 February 2019). The research adhered to the principles outlined in the Helsinki Declaration, safeguarding the human rights of participants. The methodological approach employed in the EHDLA project has been elaborated upon in a previous publication [24].

### 2.2. Adherence to the MedDiet

The Mediterranean Diet Quality Index in Children and Adolescents (KIDMED) was used to assess adherence to the MedDiet [25]. The KIDMED, created by Serra-Majem et al., is designed for individuals aged 2–24 years and focuses on the specific eating habits of the Mediterranean tradition. The questionnaire can be completed independently, through an interview, or by using a broader tool to assess eating habits, such as a food frequency questionnaire [25]. The KIDMED contains 16 yes/no questions and a final score ranging from 0–12 points. Items indicating unhealthy patterns related to the MedDiet are assigned a score of −1, whereas those reflecting healthy patterns are given +1 point. The total score from the KIDMED index is then used to classify participants into three categories: (1) optimal adherence to the MedDiet (≥8 points), (2) need for improvement to align intake with the MedDiet (4 to 7 points), or (3) very low diet quality (≤3 points) [25]. For further analyses, we folded these groups into nonoptimal adherence to the MedDiet (“very low diet quality” and “improvement needed to adjust intake to the MedDiet”) and optimal adherence to the MedDiet.

### 2.3. Food Security Status

Food security was determined by the Spanish Child Food Security Survey Module (CFSSM-S) [26]. The CFSSM-S assesses subjects’ perceptions of food security within their households, as well as their concerns about running out of food, experiencing hunger, skipping meals, reducing portion sizes, going an entire day without eating, relying solely on inexpensive foods, and being unable to maintain a balanced diet. This survey module consists of 9 items based on a 3-point Likert scale. Negative responses (i.e., “never”) are assigned 0 points, whereas positive responses (i.e., “a lot” and “sometimes”) are assigned 1 point. Following the terminology of the original study and the US Department of Agriculture [27], participants were categorized as having (1) food security (0–1 points), (2) low food security (2–5 points), or (3) very low food security (6–9 points) [24].

### 2.4. Covariates

Various factors may influence the connection between food security and adherence to the MedDiet in adolescents. When examining this relationship, it is crucial to consider multiple variables that could mediate or moderate the effects of food security. This study incorporated several covariates known to potentially impact MedDiet adherence among adolescents, including age, sex, socioeconomic status, physical activity, sleep duration, energy intake, and sedentary behavior [28,29]. Adolescents provided self-reported information on their age and sex. The Family Affluence Scale (FAS-III) was used to assess socioeconomic status [30], evaluating six items: personal bedrooms, family car and dishwasher ownership, number of bathrooms, holidays taken outside Spain in the past year, and household computers. FAS-III scores range from 0 to 13 points. The Spanish-Youth Activity Profile (YAP-S) was employed to gather data on physical activity and sedentary behavior [31], with scores calculated by summing points from each section. Energy intake was measured using a validated self-reported dietary habits questionnaire for the Spanish population [24]. Anthropometric measurements included height, measured with a portable stadiometer (Leicester Tanita HR 001, Tokyo, Japan), and body weight, measured with an electronic scale (Tanita BC-545, Tokyo, Japan). Body mass index (BMI) was calculated by dividing weight in kilograms by height in meters squared. To evaluate total sleep duration, participants were asked identical questions for weekdays and weekend days: “What time do you go to bed?” and “What time do you usually get up?”. The average daily sleep duration for each participant was determined using the formula: [(average nocturnal sleep duration on weekdays × 5) + (average nocturnal sleep duration on weekends × 2)]/7.

### 2.5. Statistical Analysis

To determine if the variables followed a normal distribution, visual techniques such as density plots and quantile-quantile (Q-Q) plots were used, in addition to the Shapiro-Wilk test. This research includes the interquartile range (IQR) and median for quantitative variables and percentages (%) and frequencies (*n*) for qualitative variables, both for those categorized by food security status and the overall sample. Generalized linear models (GLMs) were employed to explore the relationships between food security status and the MedDiet. Additionally, the 95% confidence interval (CI) and the estimated marginal mean (*M*) for the KIDMED index by food security status (food security, low food security, very low food security) were calculated. Socioeconomic status, age, energy intake, sex, physical activity, sleep duration, sedentary behavior, and BMI were included as covariates. All the statistical analyses were performed via R software (version 4.3.2) from the R Core Team in Vienna, Austria, and RStudio (version 2023.12.1 + 402) from Posit in Boston, USA, with the significance level set at *p* < 0.05.

## 3. Results

Table 1 shows the study participants’ sociodemographic, lifestyle, and anthropometric characteristics on the basis of their food security status. Among the total sample of 836 participants, 23 (2.8%) experienced very low food security, 111 (13.3%) had low food security, and 702 (84%) were food secure. The participants with low and very low food security had a median KIDMED score of 6.0 (IQR = 3.0 and IQR = 4, respectively), whereas those with food security had a median score of 7.0 (IQR = 3.0). The proportion of individuals with optimal adherence to the MedDiet (i.e., ≥8 points in the KIDMED) was 38.6% in the food security group, 35.1% in the low food security group, and 26.1% in the very low food security group.

The estimated marginal means of the KIDMED score based on the different food security statuses are shown in Figure 1. After adjusting for potential confounders, the mean KIDMED score was 7.0 (95% CI 6.7 to 7.2) in the food security group, 6.8 (95% CI 6.3 to 7.3) in the low security group, and 5.9 (95% CI 4.9 to 6.9) in the very low security group. Significant differences were identified between participants with food security and their counterparts with very low food security (*p* = 0.040). The estimated marginal means of the KIDMED score based on the CFSSM-S score are shown in Appendix A. After adjusting for all the potential confounders, we found that each additional point in the CFSSM-S score was related to a significant decrease of 0.15 points (95% CI 0.04 to 0.27) in the KIDMED (*p* = 0.006). The full results of the GLMs are shown in Appendix A (based on food security status) and Appendix A (based on the CFSSM-S score).

The predictive probabilities of having a nonoptimal adherence to the MedDiet by food security status (food security, low food security, very low food security) are shown in Figure 2. Compared with those with food security (57.2%, 95% CI 51.5% to 62.8%), a significantly greater likelihood of having nonoptimal adherence to the MedDiet was identified in adolescents with low food security (61.6%, 95% CI 50.4% to 71.8%) and in those with very low food security (77.4%, 95% CI 54.5% to 90.7%). This difference was significant only between adolescents with food security and their peers with very low food security (*p* = 0.036). The predictive probabilities of having nonoptimal adherence to the MedDiet in relation to the CFSSM-S score are displayed in Appendix A. The adjusted model revealed that for each additional point in the CFSSM-S score, there was a greater probability of having nonoptimal adherence to the MedDiet (2.6%, 95% CI 0.3% to 4.9%, *p* = 0.029). The complete results of the GLMs are available in Appendix A (according to food security status) and Appendix A (based on the CFSSM-S score).

## 4. Discussion

Our findings suggest that food security may be linked to diet quality. Specifically, compared with their peers in the food-secure group, adolescents experiencing very low food security presented significantly lower average adherence to the MedDiet and a greater likelihood of suboptimal adherence to this dietary pattern regardless of sociodemographic factor variables. Few studies have specifically examined food security and adherence to the MedDiet among children and adolescents. Nonetheless, these findings align with a study conducted among adolescents in Lebanon, which reported that lower food security status was linked with nonoptimal adherence to the MedDiet [23]. Similarly, a study of Portuguese adults revealed a significant association between low food security status and low adherence to the MedDiet [22].

Although the specific reasons for the possible relationship between the MedDiet and food security are unknown, there are possible mechanisms that may explain this relationship. On the one hand, limited access to fresh, nutrient-rich foods is a significant barrier for individuals experiencing food insecurity [20]. The economic burden required to follow the MedDiet is greater, posing a substantial obstacle for those facing food insecurity [32]. A study assessing the economic cost of the MedDiet indicated a positive correlation between the monthly cost and the degree of adherence to this eating regimen [33]. Similarly, a study comparing the costs of the MedDiet and Western dietary patterns revealed that, regardless of income, individuals who adhered more closely to the MedDiet spent more on food than those who followed a Westernized diet more closely [34]. The cost of fresh fruits and vegetables, as well as olive oil, may be one factor contributing to parents not maintaining their children’s healthy eating habits [35]. Studies indicate that a healthier diet is approximately USD 1.50 (approximately EUR 1.40) more expensive per person than a diet high in processed foods [36].

On the other hand, food insecurity can lead to irregular eating patterns due to uncertainty in food availability. Studies suggest that people experiencing food insecurity often adapt to these fluctuations by skipping meals [37] or prioritizing energy-dense products to ensure higher caloric intake during times of scarcity [38]. Identifying family members, particularly children and adolescents, who are highly vulnerable can impact their engagement with the food environment and their consumption of highly palatable foods; this, in turn, may affect their adherence to the MedDiet, which emphasizes a consistent and balanced intake of diverse fresh and healthy foods [39].

Additionally, food insecurity can be linked to heightened stress and anxiety, factors that can significantly influence food choices [40]. In this sense, individuals with chronic stress often seek energy-dense foods as a coping mechanism, making it challenging to maintain a balanced, healthy diet such as the MedDiet [41]. This tendency may partly explain the low adherence to the MedDiet, especially among adolescents, given the overlap of emotional, cognitive, and physical challenges they frequently face [42]. These circumstances are associated with increased energy intake, a diet lower in fresh foods such as vegetables, and increased obstacles to engaging in physical activity [16].

Another factor that could impact adherence to the MedDiet and food security status in adolescents is the lack of nutritional education, especially for those facing very low food security, who may not have a clear understanding of how to make healthy food choices while managing a limited budget [20]. A lack of proper nutritional education can make it difficult for adolescents and their families to make healthier dietary choices, which further reduces adherence to the MedDiet in low-food-security setting [14]. Studies have shown that very low food security status in children is associated with increased missed school attendance and reduced opportunities to complete homework assignments [43,44]. Thus, improving nutritional education could play a key role in promoting better dietary habits and increasing adherence to the MedDiet, particularly in vulnerable adolescent populations [20].

### Limitations

This research has several constraints that need to be acknowledged. Firstly, due to its cross-sectional nature, causality cannot be established; thus, future longitudinal investigations are necessary to elucidate whether decreased food security results in reduced adherence to the MedDiet. Secondly, the assessment of food security status relied on self-reported data from adolescents, potentially introducing differential bias from possible inaccuracies or overestimations. However, this issue is common in studies using self-report questionnaires. Thirdly, while the statistical analyses were adjusted for numerous variables (including sleep duration, anthropometric measurements, sex, sociodemographic factors, age, and lifestyle aspects), the possibility of residual confounding cannot be ruled out. Another limitation is the potential for selection bias, as the sample may not fully represent the broader Spanish adolescent population. Lastly, the sample in this secondary analysis is not representative of adolescents in the *Valle de Ricote*, limiting the generalizability of the results. Despite these limitations, this study offers valuable insights into the association between food insecurity and diet quality among Spanish adolescents. Moreover, the research has notable strengths, including a large sample of Spanish adolescents, a group often underrepresented in scientific studies. Additionally, the study employed validated instruments to assess both food security status and MedDiet adherence.

## 5. Conclusions

Living in a food-insecure context may compromise adherence to the MedDiet in the adolescent population. Our results remained consistent despite the consideration of other lifestyle, dietary, anthropometric, or sociodemographic variables, emphasizing the significant influence of food security status on diet quality in adolescents. In general, it is crucial to encourage healthy eating in households experiencing food insecurity, particularly for children and adolescents. Priority should be given to developing specialized, low-cost, high-nutrient programs aimed at promoting healthy eating patterns for those who are most vulnerable. Additionally, public policies must focus on improving food security to ensure that all adolescents have access to a healthy diet. Collaboration among governments, communities, and organizations can significantly influence the promotion of healthy eating habits and reduce nutritional disparities.

## Figures and Tables

**Figure 1 foods-14-00414-f001:**
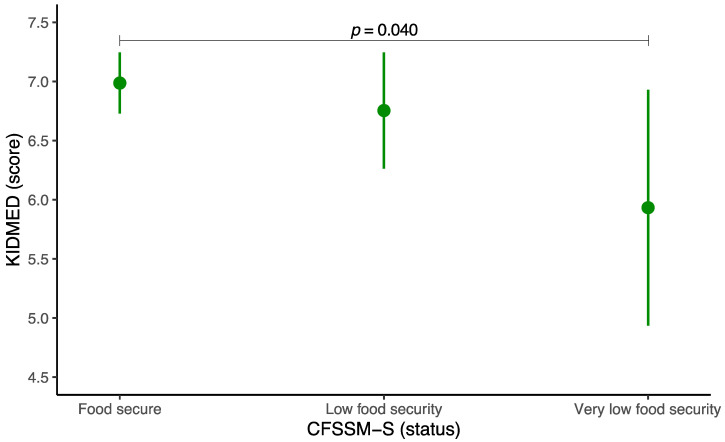
Estimated marginal means and 95% confidence intervals of the Mediterranean Diet Quality Index in children and adolescents corresponding to food security status. Socioeconomic status, energy intake, age, sex, physical activity, sleep duration, sedentary behavior, and body mass index were adjusted for. CFSSM-S, Spanish Child Food Security Survey Module. KIDMED, Mediterranean Diet Quality Index in children and adolescents.

**Figure 2 foods-14-00414-f002:**
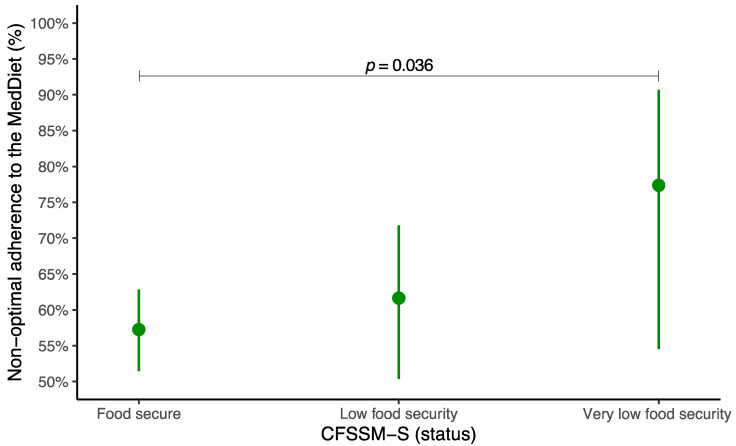
Predictive probabilities of having nonoptimal adherence to the Mediterranean diet by food security status. Socioeconomic status, energy intake, age, sex, physical activity, sleep duration, sedentary behavior, and body mass index were adjusted for. CFSSM-S, Spanish Child Food Security Survey Module.

**Table 1 foods-14-00414-t001:** Descriptive data of the study participants (*N* = 836).

Variables	Food Secure	Low Food Security	Very Low Food Security
Participants	702 (84.0)	111 (13.3)	23 (2.8)
Age (years)	14.0 (2.0)	14.0 (2.5)	15.0 (2.0)
Sex	310 (44.2)	51 (45.9)	13 (56.5)
Males			
Females	392 (55.8)	60 (54.1)	10 (43.5)
FAS-III (score)	9.0 (3.0)	7.0 (3.0)	9.0 (4.5)
YAP-S physical activity (score)	2.6 (0.8)	2.7 (0.9)	2.4 (1.0)
YAP-S sedentary behaviors (score)	2.4 (0.8)	2.6 (1.0)	2.6 (0.8)
Energy intake (kcal)	2522.1 (1398.7)	3025.5 (2323.4)	4317.3 (3759.2)
BMI (kg/m^2^)	21.6 (5.9)	22.0 (6.7)	21.8 (8.3)
Overall sleep duration (minutes)	497.1 (68.6)	492.9 (79.3)	501.4 (68.6)
KIDMED (score)	7.0 (3.0)	6.0 (3.0)	6.0 (4.0)
KIDMED (status)	271 (38.6)	39 (35.1)	6 (26.1)
Optimal			
Nonoptimal	431 (61.4)	72 (64.9)	17 (73.9)

Data expressed as median (interquartile range) for continuous variables and number (percentage) for categorical variables. BMI, body mass index; FAS-III, Family Affluence Scale-III; IQR, interquartile range; KIDMED, Mediterranean Diet Quality Index in children and adolescents; YAP-S, Spanish Youth Active Profile.

## Data Availability

The original contributions presented in the study are included in the article/Appendix A, further inquiries can be directed to the corresponding author.

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
