# Peer review of "The Role of Food Security in Mediterranean Diet Adherence Among Adolescents: Findings from the EHDLA Study"

_foods, 2025, doi:10.3390/foods14030414_

Round 1

Reviewer 1 Report

Comments and Suggestions for Authors

Aquino-Blanco et al bring a high quality paper on the relationship between food security and mediterranean diet adherence among adolescents. This study is well-rounded and presented, and does an exceptional job at representing the phenomenon with accuracy and focus on confounding factors. I have no further observation except some minor fixed, primarily regarding the study formatting:

  • Line 54 and 64 and subsequent: Please ensure that in-text citations adhere to the journal’s prescribed style. Specifically, citations should be formatted as "(1, 2)" with a space between the preceding word and the opening parenthesis.
  • Line 78: Kindly cite the original open-source publication for the referenced document to maintain accuracy and proper attribution.
  • Line 95: There appears to be a lone parenthesis that needs correction. Please address this formatting issue for clarity and consistency.

Author Response

Query 1:

Comments to the Author

  1. Line 54 and 64 and subsequent: Please ensure that in-text citations adhere to the journal’s prescribed style. Specifically, citations should be formatted as "(1, 2)" with a space between the preceding word and the opening parenthesis.

Answer 1: Thank you for your comment. The in-text citations have been revised to ensure they adhere to the journal’s prescribed style, including proper spacing and formatting.

Query 2:

  1. Line 78: Kindly cite the original open-source publication for the referenced document to maintain accuracy and proper attribution.

Answer 2: Done. Thank you.

Query 3:

  1. Line 95: There appears to be a lone parenthesis that needs correction. Please address this formatting issue for clarity and consistency.

Answer 3: Thank you for your input. The extra parenthesis after "2019 COVID-19" has been removed.

Reviewer 2 Report

Comments and Suggestions for Authors

This manuscript was interesting to read and succinct. The topic dealt with analyzed trends and connections between food security and adherence to Mediterranean dietary patterns in adolescents aged 12-17 in Spain. The manuscript is well written, the authors provided a highly detailed explanation of the statistical analyses, and performed a solid literature review of recent publications.

I have detailed some recommendations for improvement of the article below.

Abstract

General comment: it would be helpful for readers to have the age group / age range of the adolescents screened in this particular study defined in the abstract. The expected age range for adolescents can vary by country, and it would be great to have this pertinent detail highlighted in the abstract.

Introduction

Line 95: there is an extra “)” after 2019 COVID-19

Lines 99-101:This is an awkwardly written sentence and the grammar and punctuation require corrections.

Results

Table 1, line 209: Change “Boys” and “Girls” in the table to males and females. Since this paper was written with biological terminology (sex), the gender terminology should be removed.

Supplementary Files/Information

For tables S1, S2, S3 and S4, remove “Boys” and “Girls” and replace with males and females. Since your have already used sex to differentiate between males and females in the table footnotes, it is more acceptable to continue with biological terminology.

Author Response

Query 1:

Comments to the Author:

  1. Abstract: General comment: it would be helpful for readers to have the age group / age range of the adolescents screened in this particular study defined in the abstract. The expected age range for adolescents can vary by country, and it would be great to have this pertinent detail highlighted in the abstract.

Answer 1: Thank you for your suggestion. The age group of the adolescents screened in the study has been added to the abstract for clarity.

Query 2:

  1. Introduction: Line 95: there is an extra “)” after 2019 COVID-19. Lines 99-101: This is an awkwardly written sentence, and the grammar and punctuation require corrections.

Answer 2: Thank you for your feedback. The extra parenthesis after "2019 COVID-19" has been removed, and the sentence in lines 99–101 has been rewritten for improved clarity and readability.

Query 3:

  1. Results: Table 1, line 209: Change “Boys” and “Girls” in the table to males and females. Since this paper was written with biological terminology (sex), the gender terminology should be removed.

Answer 3: Done. Thank you for your input.

Query 4:

  1. Supplementary Files/Information: For tables S1, S2, S3 and S4, remove “Boys” and “Girls” and replace with males and females. Since your have already used sex to differentiate between males and females in the table footnotes, it is more acceptable to continue with biological terminology.

Answer 4: Done. Thank you.

Reviewer 3 Report

Comments and Suggestions for Authors

The manuscript submitted by Aquino-Blanco et al. can’t be considered for publication in Foods until the similarities with other published works are eliminated. It is unacceptable to have a high similarity index. The manuscript has to be rewritten to avoid plagiarism.

In the abstract, the authors should begin to provide a background statement to justify the need for conducting their research, and only then mention the study’s aims. What should be done next, future perspectives should be pointed out at the end of the abstract.

The references should be formatted accordingly, taking into consideration the journal’s guidelines.

In the Introduction, can you please provide an image of the actual Mediterranean Diet Pyramid and discuss it?

In section 2, please indicate the date of the study’s approval by the Ethics Committee.

Sections 2 and 3 should be the same (Materials and Methods).

Explain how you calculated your sample size and how it can be representative of the study population.

The study’s limitations should be discussed in a separate section and not in the Discussion.

Author Response

Query 1:

Comments to the Author:

  1. Similarity Index: The manuscript submitted by Aquino-Blanco et al. can’t be considered for publication in Foods until the similarities with other published works are eliminated. It is unacceptable to have a high similarity index. The manuscript has to be rewritten to avoid plagiarism.

Answer 1: Thank you for your feedback regarding the similarity index. The manuscript has undergone a meticulous revision to address and minimize any similarities identified in the Originality Report. A portion of the similarity arises from the necessary use of standardized terminology, methodological descriptions, and common phrases that are integral to the context of this study, particularly as it builds on the EHDLA dataset. These elements are unavoidable in ensuring the accuracy and reproducibility of our research. Additionally, the manuscript includes direct citations and properly referenced excerpts from previous studies, which are crucial for providing context, establishing a foundation for our research, and demonstrating the validity of our findings. All such references have been carefully attributed to their original sources to ensure transparency and adherence to ethical standards. We want to stress that the manuscript presents original research and novel insights within the field of food security. To further address your concern, we have rewritten sections to reduce overlap while maintaining the scientific rigor and clarity required to communicate our findings effectively.

Query 2:

  1. Abstract: In the abstract, the authors should begin to provide a background statement to justify the need for conducting their research, and only then mention the study’s aims. What should be done next, future perspectives should be pointed out at the end of the abstract.

Answer 3: Thank you for your comments. We have added background information at the beginning of the abstract to justify the need for the study.

Query 3:

  1. The references should be formatted accordingly, taking into consideration the journal’s guidelines.

Answer 3: Thank you for your comment. All references have been reviewed to ensure accuracy and compliance with the journal’s content requirements.

Query 4:

  1. Introduction: can you please provide an image of the actual Mediterranean Diet Pyramid and discuss it?

Answer 4: We have added information and discussed the updated Mediterranean Diet Pyramid. Unfortunately, we cannot provide an image of the actual pyramid in this manuscript due to copyright restrictions. We hope that the included text now provides a complete background regarding this update. Thank you.

Query 5:

  1. In section 2, please indicate the date of the study’s approval by the Ethics Committee.

Answer 5: Done. Thank you.

Query 6:

  1. Sections 2 and 3 should be the same (Materials and Methods).

Answer 6: Done. Thank you.

Query 7:

  1. Explain how you calculated your sample size and how it can be representative of the study population.

Answer 7: Thank you for your question. The original study from which this data was derived was designed to be representative of the study population for the primary outcome of interest, which was the prevalence of overweight and obesity. However, we acknowledge that this secondary analysis does not aim to be representative of the population as a whole. Instead, it focuses on a subset of the original sample, tailored to the specific objectives of this study. We have clarified this distinction in the Methods section to ensure transparency regarding the scope and limitations of the current analysis.

Query 8:

  1. The study’s limitations should be discussed in a separate section and not in the Discussion.

Answer 8: Thank you for your feedback. The limitations have now been changed to a separate section.

Round 2

Reviewer 3 Report

Comments and Suggestions for Authors

  1. Introduction: can you please provide an image of the actual Mediterranean Diet Pyramid and discuss it?

Answer 4: We have added information and discussed the updated Mediterranean Diet Pyramid. Unfortunately, we cannot provide an image of the actual pyramid in this manuscript due to copyright restrictions. We hope that the included text now provides a complete background regarding this update. Thank you.

This is not true and I encourage the authors to include the updated Mediterranean Diet Pyramid in the Introduction. You can find it in this paper: https://www.mdpi.com/1660-4601/17/23/8758. As you see it is published open access and you can incorporate it in your manuscript with adequate citation.

Author Response

Thank you for your suggestion regarding the inclusion of the updated Mediterranean Diet Pyramid in the Introduction. We appreciate the reference you provided, which is indeed an open-access publication. We have now incorporated the updated Mediterranean Diet Pyramid image into the manuscript, ensuring proper citation to comply with academic standards.

We have included the image in the manuscript. However, as per the journal's guidelines, we will await confirmation from the editorial office to ensure that including this image does not raise any legal or ethical concerns. We want to ensure full compliance with the journal's policies on the use of images.